# OS-MAP: How Far Can Computer-Using Agents Go in Breadth and Depth?

## Abstract

Computer-using agents have shown strong potential to boost human productivity and enable new application forms across platforms. While recent advances have led to usable applications, existing benchmarks fail to account for the internal task heterogeneity and the corresponding agent capabilities, as well as their alignment with actual user demands—hindering both targeted capability development and the reliable transition of research progress into practical deployment. To bridge the gap, we present OS-MAP, a benchmark for daily computer-using automation that organizes its 416 realistic tasks across 15 applications along two key dimensions: a five-level taxonomy of automation and a generalization scope derived from a real-world user demand hierarchy. To enable fine-grained analysis of required capabilities and alignment with real-world scenarios, OS-MAP evaluates agents along two dimensions: automation level across a five-level taxonomy, and generalization scope across a demand hierarchy. This design captures varying levels of required agent autonomy and generalization, forming a performance–generalization evaluation matrix for structured and comprehensive assessment. Experiments show that even State-of-the-Art agents with VLM backbones struggle with higher-level tasks involving perception, reasoning, and coordination—highlighting the need for a deeper understanding of current strengths and limitations to drive the future progress in computer-using agents research and deployment. All code, environments, baselines, and data are publicly available at https://anonymous.4open.science/r/OSMap-C2F5/.

## 1 Introduction

Computer-using agents, which can understand user intent and autonomously perform operations across digital environments, is driving the next transformation in human-computer interaction (Hu et al., 2024a;b). Powered by the extensive world knowledge, interaction capability, and tool-use abilities of Large Language Models and Vision Language Models, computer-using agents such as Operator (OpenAI, 2025), Claude 3.5 (Anthropic, 2024), UFO$^2$ (Zhang et al., 2025), and UI-TARS (Qin et al., 2025) can understand natural language instructions and interact directly with various applications in a human-like manner. Once a fixture of science fiction—like J.A.R.V.I.S. in *Iron Man*, seamlessly managing schedules, editing documents, shopping across websites, and automating routine computer tasks—such digital personal assistants are now becoming a tangible reality (Wu et al., 2024a). This transformation frees humans to focus on creative work, significantly boosting productivity and enabling new applications.

As research on computer-using agents continues to advance, an increasing number of models with strong functionalities (Qin et al., 2025; Bai et al., 2025; Xu et al., 2024a; Wu et al., 2024b) and agent systems (Jiang et al., 2025; Agashe et al., 2025; Zhang et al., 2025; Jia et al., 2024) are being proposed. Despite the rapid emergence of new methods, the open-ended semantics and diverse capability demands of computer-using tasks still hinder actual deployment. To bridge this research-to-practice gap, it is crucial to develop a principled benchmark that allows the community to quantify agent capabilities and identify specific failure points. However, existing benchmarks fall short of this goal. While spanning various platforms and scenarios, they treat tasks as flat collections, without decomposing task heterogeneity and required capabilities (Drouin et al., 2024; Xie et al., 2024; Bonatti et al., 2024; Rawles et al., 2024), making it difficult to perform fine-grained evaluation and differentiation. Moreover, task collections are typically organized around applications (Li et al.,

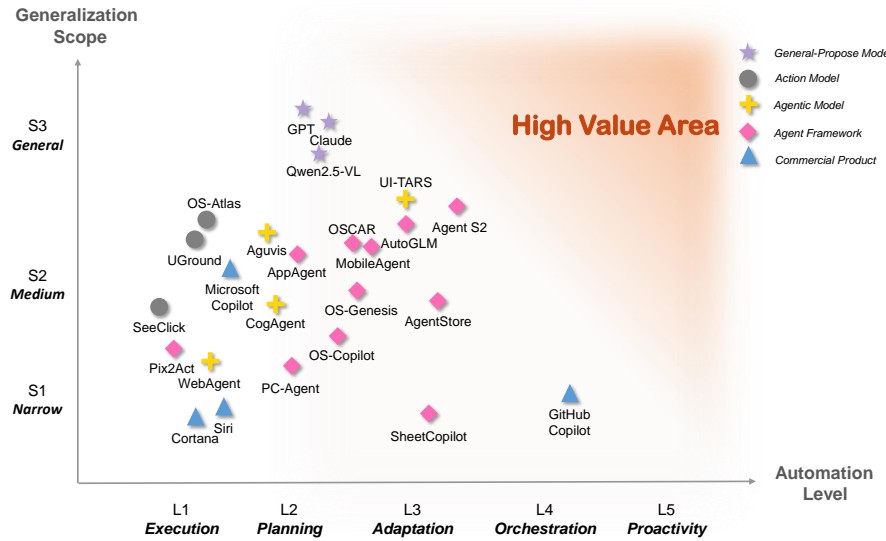

Figure 1: OS-MAP qualitative evaluation matrix, summarizing how different types of agents perform across two dimensions. General-purpose models show strong generalization, while scenario experts excel at specific tasks. Mainstream computer-using agents aim to balance both, yet still face major challenges. Agent positioning is based on reported performance, as detailed in Appendix A.

2024a; Xie et al., 2024) rather than aligned with the actual distribution of daily computer use, limiting the relevance of benchmark performance to real-world utility (Hu et al., 2024a).

To bridge these gaps, we present the OS-MAP benchmark that is grounded in dynamic desktop environments and structured along two key dimensions: **automation levels** and **generalization scopes**. First, we propose a five-level capability taxonomy based on degrees of autonomy, encompassing a wide range of computer-using tasks—from atomic execution and simple planning to disturbance adaptation, complex orchestration, and proactive behaviors. Second, we derive a real-world user demand hierarchy on daily computer-using scenarios and select representative tasks to ensure both high coverage and alignment with practical demands. Furthermore, we combine the two dimensions into a unified evaluation matrix (Figure 1), which highlights how general-purpose models, scenario experts, and mainstream computer-using agents differ in capability trade-offs between automation and generalization. The upper-right corner marks a high-value region—representing impactful yet unachieved applications—where no current agent demonstrates sufficient capability.

Across 416 tasks spanning 15 applications in OS-MAP, even the strongest existing computer-using agents achieve only an 11.5% overall success rate, with near-zero performance on higher-level tasks—falling far short of human performance. These findings underscore the importance of a principled evaluation framework. By offering both qualitative and quantitative insights into where and to what extent computer-using agents can assist humans, our framework supports comprehensive evaluation and provides a clear roadmap for future progress.

## 2 ENVIRONMENT

OS-MAP adopts and extends the OSWorld (Xie et al., 2024) infrastructure, which centers around a virtual machine (VM) and a host-side controller (VMC). This dynamic and executable environment offers fine-grained control, consistent reproducibility, flexible extensibility, and secure isolation, forming an ideal sandbox for evaluating computer-using agents in real-world scenarios.

### 2.1 TASK DEFINITION

In general, computing-using automation tasks are roughly modeled as partially observable Markov decision processes $(\mathcal{S}, \mathcal{O}, \mathcal{A}, \mathcal{T}, \mathcal{R})$. At timestep $t$, the agent resides in the environment state $s_t \in \mathcal{S}$, but only receives a partial observation $o_t \in \mathcal{O}$ (*e.g.*, the current screenshot). Based on $o_t$, the agent emits an action $a_t \in \mathcal{A}$ (*e.g.*, a structured text `click(350,600)`). The environment transitions to the next state $s_{t+1} = \mathcal{T}(s_t, a_t)$ via the transition function $\mathcal{T}$, which is governed by the underlying

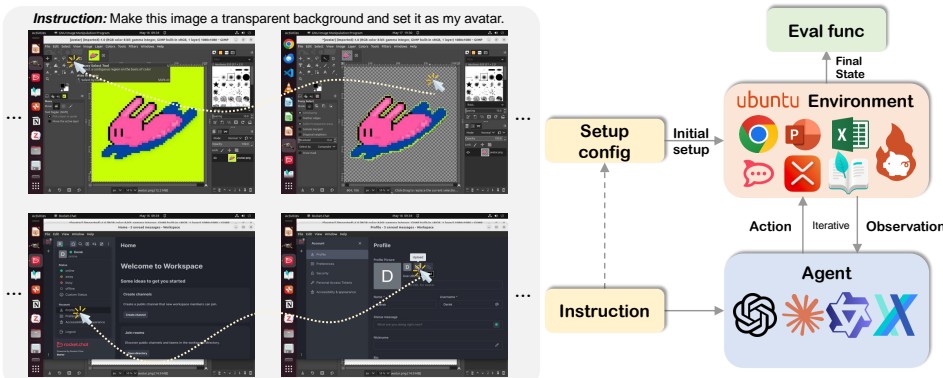

Figure 2: OS-MAP is built on an executable desktop environment designed for daily computer tasks, integrating a suite of applications and tools. It provides the infrastructure for reliable evaluation by handling task initialization and success verification. Agents interact autonomously via GUI operations, guided by instructions and screenshot perception.

software and OS logic, revealing a new observation $o_{t+1}$. This process continues iteratively until the agent actively issues a terminal action (*i.e.*, DONE or FAIL), or passively exceeds a predefined step limit. After termination, the system determines whether the task is successfully completed and provides a final outcome reward $r \in \mathcal{R} = \{0, 1\}$, without any intermediate process rewards.

## 2.2 ENVIRONMENT STRUCTURE

**Task lifetime.** Each task is specified by a JSON file defining initialization, instruction, and evaluation protocols. As shown in Figure 2, evaluation begins by restoring a designated VM snapshot and running lightweight setup routines. The agent then enters the interaction loop, receiving observations from and sending actions to the VM via the VMC. This loop continues until the agent terminates the episode, either actively or passively. The evaluator then compares the VM state to reference criteria and returns a binary reward. See Appendix B.1 for details.

**Initialization and evaluation configuration.** Task setup in OS-MAP combines VM snapshots with modular configuration functions, supporting scalable and flexible task creation. Standard initialization adopt reusable OSWorld functions (*e.g.*, file downloading, shell commands), while more complex setups—such as software installation or database configuration—are manually performed and captured as directly restorable snapshots. Evaluation integrates both state-based and action-based assessments to support tasks with varying levels of automation. Depending on the task, state evaluation may involve file comparison and system state inspection, or execution-based verification. See Appendix B.2 and B.3 for detailed initialization and evaluation modes.

**Observation and action space.** Recent computer-using agents research increasingly gravitates toward human-like interaction paradigms: raw pixel screenshots as observations and atomic keyboard-/mouse operations as actions. OS-MAP adopts this design, using raw screenshots as input—without accessibility trees or Set-of-Marks (Yang et al., 2023) annotations—for simplicity and broader applicability. The action space follows OSWorld's 13 atomic operations and 3 meta-actions (*i.t.*, WAIT, FAIL, DONE). See Appendix B.4 and B.5 for detailed space descriptions with examples.

## 3 BENCHMARK

OS-MAP comprises 416 real-world computer-using automation tasks on 15 Ubuntu applications, spanning diverse everyday scenarios. Tasks are categorized along two orthogonal dimensions: **automation level**, capturing the degree of agent autonomy, and **generalization scope**, defined by a hierarchical demand taxonomy, measuring agents' capability transferability. Together, they form a structured **evaluation matrix** (Fig. 1) supports systematic evaluation. The following sections introduces the automation levels (§3.1), generalization scope (§3.2), evaluation matrix (§3.3), task curation pipeline (§3.4), and benchmark statistics (§3.5).

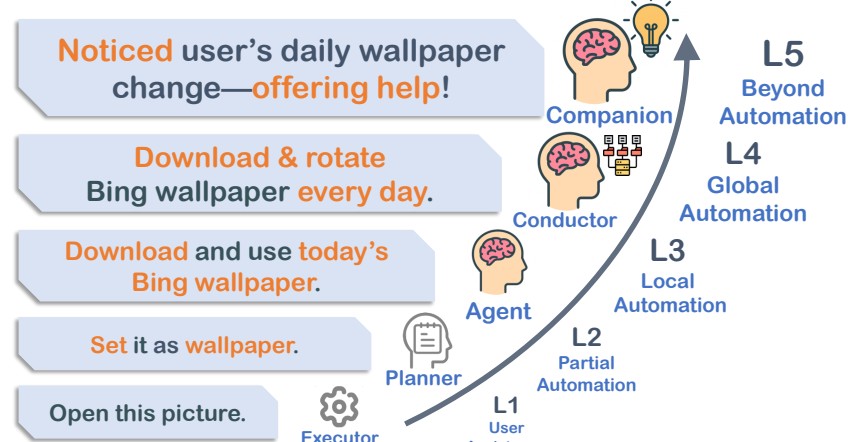

Figure 3: Automation levels demonstration on a specific task: rotating wallpapers daily. From the user's perspective, achieving the same goal involves increasing agent responsibility and decreasing user involvement as automation level rises. Task executions become longer and more complex, reflecting the shifting division of labor between human and the agent.

### 3.1 AUTOMATION LEVELS

Real-world computer automation varies in task complexity, user involvement, and agent responsibility. To support consistent evaluation across these variations, we introduce a five-level automation taxonomy, inspired by SAE driving automation taxonomy (Committee, 2021) and grounded in the division of labor between humans and agents. Figure 3 illustrates how a concrete task manifests across all five interaction modes. Each level reflects a specific degree of autonomy in planning and execution, shaped by both task complexity and the expected user role.

**L1: Reactive executor.** The agent executes user-defined atomic operations (e.g., clicks, keystrokes) without making decisions. Task planning remains entirely user-driven. This stage primarily evaluates perceptual grounding and command-to-action mapping—capabilities that many grounding models (Cheng et al., 2024; Wu et al., 2024b; Gou et al., 2024) specifically target.

**L2: Deterministic planner.** The user specifies only the task goal, leaving the agent to autonomously plan and execute actions under ideal and predictable conditions. High-level task decomposition remains user-driven, and intervention is required when failures occur. This stage tests prior knowledge and basic planning, representing the operational level of most current agents and proprietary models.

**L3: Adaptive agent.** L3 emphasizes robustness in dynamic, noisy, and partially observable environments. Agents must adapt plans autonomously in response to unpredictable events or evolving interface states. While users still define high-level goals, they no longer need to monitor or intervene during execution. Only a small subset of agents specifically designed for adaptivity reach this level, demonstrating resilience and flexible subtask completion under real-world conditions.

**L4: Global conductor.** The agents take full responsibility for decomposing high-level goals and orchestrating complex workflows involving subgoals, cross-application context switching, and tool usage. Acting as autonomous top-level orchestrators, they coordinate entire tasks end-to-end, with users only issuing goals and verifying outcomes. As shown in our results, no current agent effectively handles this level, though emerging multi-agent approaches show promise.

**L5: Proactive companion.** L5 marks a shift from reactive execution to proactive collaboration. The agent continuously monitors context, anticipates user needs, and initiates helpful actions without explicit instructions. It learns from long-term interactions to provide personalized support as an intelligent digital companion. While still an underexplored concept, with only a few studies across different scenarios (Lu et al., 2024; Chen et al., 2025; Chaves and Gerosa, 2021; Liao et al., 2023), it holds significant promise for future applications. OS-MAP does not yet include L5 tasks and we leave them for future improvements.

## 3.2 GENERALIZATION SCOPE

While §3.1 focuses on structural organization via automation levels, this section turns to the content dimension. Designing meaningful tasks for computer-using agents is challenging. Prior benchmarks often rely on predefined applications sets (Xie et al., 2024; Bonatti et al., 2024; Li et al., 2024a; Chai et al., 2025) or template-based generation (Rawles et al., 2024; Drouin et al., 2024). In contrast, we take a demand-driven approach—identifying common daily use cases and deriving tasks accordingly—to ensure realism, representativeness, and practical relevance.

**Demand hierarchy.** We define a three-level hierarchy: domains, scenarios, and representative tasks with applications, guided by industry data (Tower, 2025) and public surveys (OECD, 2025). Starting from the *State of Mobile 2025* report, we adapt mobile usage statistics to the desktop setting by excluding mobile-specific categories (*e.g.*, payments) and adding desktop-relevant ones (*e.g.*, office work), forming six domains: work, study, life services, entertainment, creative production, and system management. Figure 4 illustrates this demand hierarchy details. Scenarios are derived by aligning app subcategories with activity metadata from OECD *ICT Access and Usage Database* (OECD, 2025). Tasks are then selected through expert review and LLM-assisted ideation, filtered by clarity, reproducibility, and independence from real-world accounts or network-side effects.

**Generalization scope.** Anchored in this hierarchy, we define three scopes of generalization: S1 (Narrow), S2 (Domain-Level), and S3 (General) to characterize agents' capability breadth across diverse user demand. An S1 agent handles tasks within a single scenario (*e.g.*, calendar management). An S2 agent succeeds across multiple scenarios within a domain (e.g., document editing, emailing, and scheduling in the work domain). An S3 agent demonstrates S2-level performance across most or all six domains, effectively acting as a cross-domain generalist for daily computer-using assistance.

## 3.3 EVALUATION MATRIX

We further integrate the two orthogonal dimensions—automation levels (L1–L5) and generalization scopes (S1–S3)—into a two-dimensional evaluation matrix, enabling a systematic assessment of both the depth and breadth of agent capabilities, as presented in Figure 1. This depth–breadth perspective aligns with the *performance–generality* framework proposed in earlier AGI research (Morris et al., 2024; Zhang et al., 2024a). In the context of computer-using, **performance** denotes the extent to which an agent can operate independently from human intervention within collaborative settings, as reflected by the task complexity across the automation levels. **Generality** refers to the range of tasks where the agent meets the performance threshold, anchored in its coverage of the demand hierarchy.

By decoupling performance and generality, the matrix provides a fine-grained evaluation of CUAs' practical utility by revealing strengths and limitations and supporting clear comparisons across systems with differing design priorities. The structure also scales naturally—new tasks or scenarios can be added to underexplored regions without disrupting the overall framework. Most importantly, it offers a clear developmental roadmap, guiding researchers and practitioners in setting progressive goals along both dimensions toward building more capable and general-purpose agents.

## 3.4 TASK CURATION PROCESS

Each task in OS-MAP is created following a standardized six-step process grounded in the two-dimensional organization framework: (1) task selection, (2) exploration and specification, (3) instruction and configuration, (4) reference state preparation, (5) evaluation setup, and (6) cross-validation. Detailed descriptions of each step are provided in Appendix C.1. To ensure correctness and stability in an open-ended environment, each stage of this process demands significant manual effort and verification. Appendix C.2 presents the full design and refinement process of a representative L4 task. We also incorporate and adapt OSWorld tasks by mapping them to our difficulty levels and generalization tiers. All included tasks undergo the same validation pipeline, while those with invalid formats or ambiguous feasibility are excluded. Related details are discussed in Appendix C.3.

## 3.5 BENCHMARK STATISTICS

**Statistics.** Figure 4 presents the distribution of tasks in OS-MAP across the user-centered demand hierarchy. Based on industry surveys (Tower, 2025; OECD, 2025), we define a three-level demand hierarchy comprising 6 top-level needs, 18 sub-needs, and 45 concrete scenarios, spanning 15 representative applications and covering a broad range of daily computer-using situations. In total,

Figure 4: Task distribution on the demand hierarchy in OS-Map benchmark.

Table 1: Statistics of OS-Map.

| Task Type | Statistics |
|---|---|
| **Total Tasks** | **416 (100%)** |
| - Single-App | 283 (62.3%) |
| - Multi-App | 154 (37.7%) |
| **Automation Level** | |
| - L1: Execution | 25 (6.0%) |
| - L2: Planning | 234 (56.3%) |
| - L3: Adaptability | 115 (27.6%) |
| - L4: Orchestration | 42 (10.1%) |
| **Source** | |
| - Authors | 161 (38.7%) |
| - Labeled from OSWorld | 255 (61.3%) |
| Avg. Words of Task Instructions | 34.3 |
| Avg. Steps | 11.4 |

OS-Map contains 416 tasks representative of their respective scenarios. Among them, 138 tasks are meticulously designed by the authors, while the remaining 296 are relabeled and filtered for ambiguity and redundancy from OSWorld (Xie et al., 2024) as Appendix C.3 describes. Table 1 provides more detailed statistics. Notably, 37.7% of tasks involve multi-app workflows, posing significant challenges to agents' adaptation and orchestration capabilities.

**Comparison with existing benchmarks.** Table 2 compares OS-Map with existing efforts across key dimensions. First, OS-Map builds on an executable environment, inheriting the architecture, utility functions, and evaluation tools from OSWorld (Xie et al., 2024). This ensures controllability and flexible open-domain scalability. Second, we expand the number of applications and tasks, including a substantial portion of cross-application tasks, thereby enhancing the task diversity. Most importantly, we introduce a fine-grained evaluation framework based on both task difficulty levels and user demand hierarchy. These two axes are further integrated into a structured, two-dimensional evaluation matrix, enabling systematic, detailed comparisons and offering clear guidance for future development—an aspect largely overlooked by existing benchmarks.

Table 2: Comparison of different environments for benchmarking CUAs. The columns indicate: dynamic executable environment provided (Exec. Env.?), the ease of adding new tasks involving arbitrary applications in open domains (Scal. Env.), the number of applications or websites (#Apps/sites), the number of task instances and templates (if applicable) (# Inst. (# Temp.)), inclusion of cross-app tasks (Cross-app?), whether to provide evaluation based on task difficulty (Task Diff. Levels?), demand perspective (Demand Scope?), or a multi-dimensional structure (Struct. Eval.?).

| Benchmark | Exec. Env.? | Scal. Env.? | # Apps/ sites | # Inst. (# Temp.) | Cross-app? | Task Diff. Levels? | Demand Scope? | Struct. Eval.? |
|---|---|---|---|---|---|---|---|---|
| GAIA Mialon et al. (2023) | ✗ | - | - | 466 | ✗ | ✓ | ✗ | ✗ |
| Mind2Web Deng et al. (2023) | ✗ | - | 137 | 2350 | ✗ | ✗ | ✗ | ✗ |
| WebVoyager He et al. (2024) | ✗ | - | 15 | 643 | ✗ | ✗ | ✗ | ✗ |
| PixelHelp Li et al. (2020) | ✗ | - | 4 | 187 | ✗ | ✗ | ✗ | ✗ |
| AitW Rawles et al. (2024) | ✗ | - | 357+ | 30k | ✗ | ✗ | ✗ | ✗ |
| OmniAct Kapoor et al. (2024) | ✗ | - | 60+ | 9802 | ✗ | ✗ | ✗ | ✗ |
| WebShop Yao et al. (2022) | ✓ | ✗ | 1 | 12k (1) | ✗ | ✗ | ✗ | ✗ |
| WebArena Zhou et al. (2023) | ✓ | ✗ | 6 | 812 (241) | ✗ | ✗ | ✗ | ✗ |
| WorkArena Drouin et al. (2024) | ✓ | ✗ | 1 | 23k (29) | ✗ | ✗ | ✗ | ✗ |
| AndroidArena Xing et al. (2024) | ✓ | ✗ | 13 | 221 | ✓ | ✓ | ✗ | ✓ |
| AndroidWorld Rawles et al. (2024) | ✓ | ✓ | 20 | ∞ (116) | ✓ | ✗ | ✗ | ✗ |
| AndroidAgentArena Chai et al. (2025) | ✓ | ✓ | 21 | 201 | ✗ | ✓ | ✗ | ✗ |
| OSWorld Xie et al. (2024) | ✓ | ✓ | 9 | 369 | ✓ | ✗ | ✗ | ✗ |
| Spider2-V Cao et al. (2024) | ✓ | ✓ | 20 | 494 | ✓ | ✗ | ✗ | ✗ |
| WindowsAgentArena Bonatti et al. (2024) | ✓ | ✓ | 11 | 154 | ✓ | ✗ | ✗ | ✗ |
| TheAgentCompany Xu et al. (2024b) | ✓ | ✗ | 6 | 175 | ✓ | ✗ | ✓ | ✗ |
| ScienceBoard Sun et al. (2025) | ✓ | ✓ | 6 | 169 | ✓ | ✓ | ✗ | ✓ |
| OS-Map | ✓ | ✓ | 15 | 416 | ✓ | ✓ | ✓ | ✓ |

## 4 EXPERIMENTS AND ANALYSIS

### 4.1 EXPERIMENTAL SETTINGS

**Agent types.**    We construct three types of computer-using agents based on different types of state-of-the-art models: **(1) General baselines**: directly uses a general-purpose VLMs (`GPT-4o` (Hurst et al., 2024), `Claude-3.7-Sonnet` (Anthropic, 2025), `Gemini-2.5-Pro` (Team, 2025), `Qwen2.5-VL-72B` (Bai et al., 2025), `InternVL3-8B` (Zhu et al., 2025)) to perform each task end-to-end. **(2) GUI-specific model baseline**: executes tasks end-to-end using GUI-specialized VLMs (`UI-TARS-72B` (Qin et al., 2025)). **(3) Planning-Grounding**: to compensate for the imprecise grounding abilities of general models, `GPT-4o` (Hurst et al., 2024) is used to conduct high-level plans, which are then refined by lightweight GUI action models (`Aguvis-7B` (Xu et al., 2024a), `OS-ATLAS-Base-7B` (Wu et al., 2024b), `UGround-7B` (Gou et al., 2024), `GUI-Actor-7B` (Wu et al., 2025)) for precise grounding.

**Agent settings.**    All three agent types share a common decision-making and interaction pattern, along with similar prompting strategies. Specifically, the agent interacts with the environment under the guidance of a system prompt, which includes descriptions of the task goal, observation space, action space, and required output format. At each step, the agent generates an action based on the current screenshot and the three most recent rounds of interaction history. Detailed prompts and interaction protocols are provided in Appendix E.

### 4.2 RESULTS

We compare the performance of the above four computer-using agents types powered by different models on OS-MAP, as presented in Table 3. We summarize our key empirical results as follows:

**Computer-using agents remain far from practical deployment.**    Despite recent advances, current agents exhibit consistently poor performance across all levels of automation, with many near zero, highlighting a substantial performance gap from human users. This suggests that existing models still struggle with core capabilities such as grounding.

**Agents' performance exhibits a stepwise decline across automation levels.**    Among the evaluated models, `UI-TARS-72B` achieves the best balance of visual grounding, robust planning, and task generalization, significantly outperforming other competitors. It performs well when tasks include step-level guidance (L1) and maintains solid performance on basic planning (L2). However, its advantage drops markedly on environmental adaptation (L3) and multi-context orchestration (L4), suggesting that **adaptive reasoning and long-horizon planning remain key challenges**.

**Open-source models have achieved competitive end-to-end performance.**    Although smaller in scale, open-source models fine-tuned on GUI-specific data (Bai et al., 2025) or trained in GUI-centric environments (Qin et al., 2025) demonstrate better performance than proprietary general-purpose models in end-to-end execution. Their superiority stems from targeted training in GUI contexts, which enhances planning stability and task adaptation in complex desktop environments.

**Tailored training and agentic setting yield better computering-using performance.**    Compared to general models, GUI-specific models (Qin et al., 2025) and interleaved planning-grounding agents (Xu et al., 2024a; Wu et al., 2024b; Gou et al., 2024; Wu et al., 2025) gain a significant improvement. The specially designed GUI training makes models familiar with computer environments, while the planning-grounding agents combine the world knowledge and strategic planning of general models with the precise perception and control ability of the GUI-oriented models.

### 4.3 ANALYSIS

To understand key challenges behind poor performance, this section analyzes representative failure cases to uncover core factors that lead to agent breakdowns. We highlight both general capability gaps observed across agents and level-specific bottlenecks tied to increasing automation levels. These insights shed light on where agents fall short and inform more targeted future improvements. See Appendix F for a more detailed case analysis with screenshots.

Table 3: Success rates of computer-using agents on OS-MAP. We present each agent backbone's performance on tasks across different automation levels. Proprietary VLMs , and Open-Source VLMs are distinguished by color. In Planning-Grounding setting, `GPT-4o` is used as the planning model.

| Agent Type | Model | Success Rate (↑) | | | | |
|---|---|---|---|---|---|---|
| | | L1 | L2 | L3 | L4 | Overall |
| | `GPT-4o` | 12.0% | 1.3% | 1.7% | 0.0% | 1.9% |
| | `Claude-3.7-Sonnet` | 0.0% | 3.8% | 0.0% | 0.0% | 2.1% |
| General Baselines | `Gemini-2.5-Pro` | 8.0% | 10.6% | 2.7% | 2.4% | 7.5% |
| | `Qwen2.5-VL-72B` | 32.0% | 7.9% | 1.0% | 0.0% | 6.6% |
| | `InternVL3-8B` | 8.0% | 1.6% | 1.0% | 0.0% | 1.6% |
| GUI-Specific Baseline | `UI-TARS-72B` | 48.0% | 14.0% | 1.0% | 0.0% | 11.4% |
| | `Aguvis-7B` | 4.0% | 4.7% | 1.8% | 0.0% | 3.4% |
| Planning-Grounding | `OS-ATLAS-Base-7B` | 8.0% | 6.4% | 1.8% | 0.0% | 4.6% |
| | `UGround-7B` | 16.0% | 4.6% | 1.8% | 0.0% | 4.0% |
| | `GUI-Actor-7B` | 40.0% | 15.1% | 1.8% | 0.0% | 11.5% |
| Human Performance | | 96.0% | 74.8% | 65.2% | 59.5% | 71.9% |

### 4.3.1 GENERAL FAILURES

**Poor instruction following.** This manifests as frequent violations of the required output format. A typical case is `Claude-3.7-Sonnet` issuing an `OPEN_FILE_EXPLORER` action when openning the file manager—despite the valid action space only contains atomic mouse and keyboard operations.

**Severe hallucination.** Due to limited perception and reasoning, agents often wrongly assume that previous actions have succeeded, and occasionally exhibit drastic hallucinations—*e.g.*, mistaking the activities window for Chrome and attempting to search within it (Figure 13 in Appendix F).

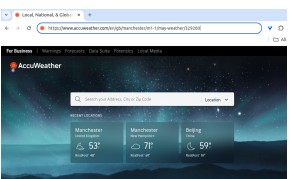

Figure 5: Agent prefers entering a URL instead of navigating websites.

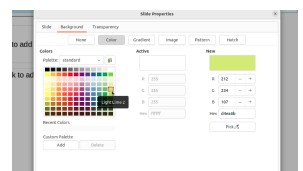

Figure 6: GUI action model fails in the grounding of the green block.

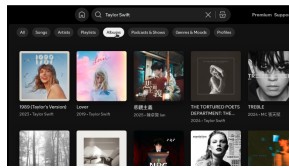

Figure 7: Searching for the album Taylor Swift instead of albums *by* Taylor Swift.

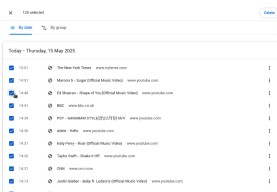

Figure 8: Agent is deleting all history, not just those from YouTube.

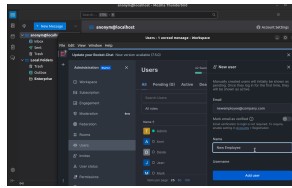

Figure 9: Agent start filling out the form before clarifying information.

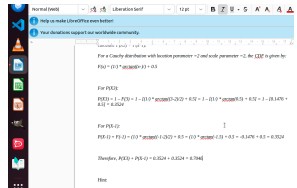

Figure 10: Calculating calculus internally instead of trying tools in the context.

Figure 11: Failure cases of each automation levels, reflecting bottlenecks in core capabilities.

### 4.3.2 LEVEL-WISE BOTTLENECKS

**L1: execution.** Proprietary models exhibit poor grounding capabilities, often preferring command-line operations or direct URL jumps (Figure 5). In contrast, GUI action models demonstrate more human-aligned interaction patterns, yet still struggle with locating non-textual elements (Figure 6).

**L2: planning.** Agents are prone to two common failure modes at this level: **(1) distraction by similar but incorrect options**—for example, searching for the album Taylor Swift instead of *all albums by Taylor Swift* (Figure 7); and (2) **neglecting specific task constraints**—*e.g.*, deleting all browsing history instead of only entries related to YouTube (Figure 8).

**L3: adaptability.** Agents demonstrate basic proactive exploration (*e.g.*, inspecting potential directories before file operations) and reactive handling (*e.g.*, closing unexpected pop-ups). However, they **struggle with fallback strategies under deviation**, such as failing to exit full-screen mode via hotkeys (Figure 14 in Appendix F), or activating theater mode before resizing, which hides the required controls (Figure 15 ). They also show **poor awareness of implicit task context** (Figure 16).

**L4: orchestration.** Agents exhibit major bottlenecks in all challenges including goal decomposition, dependency tracking, context switching, and tool use: **unclear decomposition** leads to aimless clicking (Figure 17 in Appendix F); **misordered context switches** break task dependencies (Figure 9) or initiating transactions before checking the balance (Figure 18); and **failure to leverage external tools** (Figure 10).

## 5 RELATED WORK

**Computer use benchmarks.** Existing computer-using benchmarks can be broadly categorized along several dimensions: by platform (*e.g.*, Web (Deng et al., 2023; Zhou et al., 2023), Desktop (Xie et al., 2024; Cao et al., 2024; Xu et al., 2024b), or Mobile (Rawles et al., 2023; 2024; Chai et al., 2025)); by task type (*e.g.*, understanding (Liu et al., 2024; Chen et al., 2024a), grounding (Cheng et al., 2024; Nayak et al., 2025), and end-to-end automation); and by scenario domain (*e.g.*, everyday, office (Drouin et al., 2024), or professional (Cao et al., 2024; Li et al., 2025)). A recent trend is the adoption of dynamic environments (Xie et al., 2024; Cao et al., 2024; Xu et al., 2024b; Rawles et al., 2024; Sun et al., 2025). Focusing on end-to-end evaluation in daily scenarios on a dynamic desktop environment, OS-MAP is the first to systematically analyze task structures and automation levels grounded in real-world user needs, bridging capability evaluation with practical relevance.

**Computer-using agents.** Recent advances in computer-using agents have been highly diverse. For modeling, efforts have focused on enhancing visual perception through high-resolution (Hong et al., 2024; Li et al., 2024b) or adaptive cropping and token selection (Zhang et al., 2024b; Lin et al., 2024; Wu et al., 2025) techniques. For data, two trends have emerged: (1) large-scale multi-task web-based pretraining (Cheng et al., 2024; You et al., 2024; Chen et al., 2024b; Wu et al., 2024b; Gou et al., 2024; Qin et al., 2025), and (2) supervised fine-tuning on high-quality interaction trajectories (Zhang et al., 2024c; Sun et al., 2024; Su et al., 2025). Reinforcement learning has been introduced to improve error recovery, and long-horizon reasoning (Fan et al., 2025; Lu et al., 2025; Xia and Luo, 2025; Liu et al., 2025). A parallel line of work builds ReAct-style (Yao et al., 2023) agents coordinating structured functional modules, with growing emphasis on hierarchical planning, systematic memory organization, and collaborative multi-agent systems (Agashe et al., 2024; Wu et al., 2024a; Jia et al., 2024; Agashe et al., 2025; Jiang et al., 2025; Wang et al., 2025; Zhang et al., 2025).

**AI capability levels.** Both industry and academia have long explored ways to define graded AI capabilities (Sheridan and Parasuraman, 2005; Parasuraman et al., 2000; Goertzel, 2014). A well-known example is the levels of driving automation (Committee, 2021) based on the human–system driving collaboration model. Recently, researchers have proposed grading schemes for artificial general intelligence with the performance-generality capability framework (Morris et al., 2024; Zhang et al., 2024a). A related research (Li et al., 2024c) discusses the intelligence levels of personal LLM agents in terms of collaboration patterns, but with abstract classification, vague levels, and overlapping capabilities. Our work grounds the performance–generality perspective in the concrete domain of computer automation, introducing clear levels aligned with real-world tasks, and further instantiates this taxonomy as OS-MAP benchmark to support quantitative evaluation.

## 6 CONCLUSION

In this work, we propose a two-dimensional evaluation framework for computer-using agents, spanning automation levels and generalization scopes. We instantiate it as the OS-MAP Benchmark, comprising 416 tasks across 15 desktop applications, executed in a controllable and extensible environment to ensure quantitative and reproducible evaluation. Despite recent progress, OS-MAP remains highly challenging—state-of-the-art general-purpose and GUI-specialized VLMs still fall far short of human performance. Through in-depth failure analysis, we identify key capability bottlenecks at each automation level, laying a foundation for targeted improvements in future research.

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

## A  Qualitative Evaluation Matrix

This section explains the qualitative criteria used to position each method in the evaluation matrix (Figure 1). Given the subjective and heuristic nature of this analysis, the capability levels shown are approximate and do not reflect strict objectivity or fine-grained scale.

- **Academic methods.** For research models, capability levels are estimated based on their task scope, qualitative behavior, and whether they address key challenges or demonstrate core abilities. Quantitative results on relevant benchmarks are then used to refine their positions. For example, SeeClick Cheng et al. (2024), U-Ground Gou et al. (2024), and OS-Atlas Wu et al. (2024b) are all evaluated on the ScreenSpot Cheng et al. (2024) dataset, which focuses on GUI action grounding—a task category near L1. The latter two models incorporate rudimentary planning and can independently complete simple end-to-end tasks, suggesting capabilities closer to L2. Their exact placement is further adjusted based on performance scores (*e.g.*, overall accuracy) and domain generalization, as ScreenSpot includes multiple domains. Other methods are evaluated similarly, using additional benchmarks such as AITW Rawles et al. (2023), GAIA Mialon et al. (2023), WebArena Zhou et al. (2023), and OSWorld Xie et al. (2024).

- **General-purpose models.** For models like GPT, we base our assessment on their qualitative and quantitative performance in OSWorld Xie et al. (2024) and OS-Map, under both end-to-end and planning-grounding settings. These models show strong generalization—able to plan in nearly any scenario—but limited adaptivity, often struggling with unexpected events or common errors. As a result, they are positioned in the upper-middle region of the matrix.

- **Commercial products.** For tools like Microsoft Copilot, which lack quantitative evaluations or OSWorld-style experiments, we rely on a combination of official capability descriptions, public user discussions, and authors' own usage experience. Earlier commercial products like Siri offer narrow functionality and low automation, while GitHub Copilot shows high-level code generation capabilities, often anticipating user needs. Microsoft Copilot for `Windows 11` provides a more balanced and moderate level of capability and coverage.

## B  Environment Structure

The core of the OS-Map environment consists of a virtual machine (VM) and a virtual machine controller (VMC). The host machine runs a VM using virtualization software such as VMware. This VM serves both as the source of visual observations and the target for action execution by the agents. The host communicates with the VM through a virtual network, enabling initialization, observation extraction, file transfer, and other forms of control. These components, together with tools for launching the VM, loading snapshots, and managing execution states, collectively form the VMC, which runs on the host side. The following sections detail how this architecture supports the task lifetime, initialization and configuration, state-based evaluation, and the design of the observation and action spaces.

### B.1  Task Lifetime

OS-Map consists of 416 tasks across diverse scenarios, each controlled and executed sequentially by a main evaluation loop. For each iteration, a new task evaluation is initiated. The lifetime of a task is composed of the following five stages:

1. Initialization. To ensure reproducibility, each task begins by loading a designated snapshot. Afterward, a predefined initialization script is executed. Snapshots and initialization scripts are designed to work in tandem, offering both high flexibility and low initialization overhead.

2. Task execution. Once initialized, the system enters the execution loop. At each step, the VMC captures the current observation and passes it to the agent. Based on the current state and interaction history, the agent outputs a textual action. This action is parsed and executed by the VMC within the VM. The loop continues until the agent either terminates voluntarily (via `DONE` or `FAIL`) or reaches the maximum allowed number of steps.

3. Post-execution configuration (optional). For certain tasks, the final state after agent execution is not directly extractable. In such cases, additional actions are required to bring the system into a

verifiable state. For example, after adding an item to the cart on the Decathlon website, the system needs to open the cart page so the evaluator can verify the result by inspecting the DOM tree.

4. State extraction. The VMC includes a set of state extraction functions designed to retrieve relevant information from the VM. These serve as input for the next evaluation step.

5. Evaluation. Evaluation functions are task-specific and compare the extracted state against expected conditions. Depending on the nature of the state, corresponding comparison logic is applied—such as string matching, file equivalence, or key–value comparison.

### B.2 INITIALIZATION CONFIGURATIONS

Task initialization in OS-MAP relies on a combination of restorable VM snapshots and configuration scripts. For simple tasks, initialization can be performed directly via scripts, which are pre-written command sequences that encapsulate commonly used operations—such as file downloads, application launches, API calls, webpage interactions via Playwright, and shell commands. For more complex or customized tasks (e.g., pre-created users and messages in Rocket.Chat), manual setup is conducted in advance and saved as a snapshot for fast recovery. In practice, task initialization uses both snapshot recovery and lightweight runtime configuration: snapshots (either from leaf nodes or key intermediate nodes of the snapshot tree) are loaded first, followed by scripted configuration. This hybrid approach ensures high flexibility and minimizes initialization time.

### B.3 STATE-BASED EVALUATION

The primary motivation for introducing a dynamic environment is to enable state-based evaluation. The underlying logic is that as long as the system ultimately reaches a predefined desired state, the task is considered successfully completed—regardless of the specific sequence of actions taken to reach that state. This approach allows for a fair comparison between different execution trajectories of the same task.

Accordingly, each task JSON file must define both the target state and the method for extracting relevant system states. Common examples include retrieving specific files, reading software or system configurations, or extracting the content of rendered webpages via Playwright. In certain tasks, a post-config step is required to convert hard-to-access intermediate states into more easily extractable forms. Implementing state-based evaluation requires substantial reverse engineering of software and operating systems to locate and extract relevant data. Once the VM's state is extracted to the host machine, an evaluation function compares it against the target state to determine whether the task has been successfully completed.

The evaluation methods are tailored to the extracted state types, typically involving file comparisons or configuration matching. In some cases, more specialized metrics—such as image similarity or fuzzy text matching scores—are used. All evaluation results are ultimately converted into a binary outcome, indicating task success or failure.

### B.4 OBSERVATION SPACE

OS-MAP use screenshot for the only observation modality. Following OSWorld Xie et al. (2024), the VMC takes full-screen screenshots and preserves the cursor to align with human perception of the UI. The default resolution is $1920 \times 1080$, and supports adjustments to avoid overfitting on absolute pixel coordinates and generalization studies.

While previous benchmarks Xie et al. (2024); Cao et al. (2024) also used inputs such as the accessibility (a11y) tree and Set-of-Marks Yang et al. (2023) (SoM) prompted screenshots, OS-MAP relies solely on screenshots for two main reasons:

- Screenshots are easy to capture and align closely with human perception.
- They preserve rich visual information, whereas a11y trees and SoM formats can be overly excessively verbose, lossy, inaccurate, or unavailable in visually complex interfaces.

Although structured inputs sometimes yield better performance, recent methods have increasingly shifted toward using VLMs on raw screenshots Qin et al. (2025); Bai et al. (2025); Wu et al. (2024b), making pure visual input the more general and future-proof approach.

Table 4: Action types and parameters defined in action space COMPUTER_13, a variance we created for the potential reinforcement learning research based on our environment.

| Action Type | Parameters | Note |
|---|---|---|
| MOVE_TO | x, y | Move the cursor to the specified position |
| CLICK | button, x, y, num_clicks | Click the left button if the button not specified, otherwise click the specified button; click at the current position if x and y are not specified, otherwise click at the specified position |
| MOUSE_DOWN | button | Press the left button if the button not specified, otherwise press the specified button |
| MOUSE_UP | button | Release the left button if the button not specified, otherwise release the specified button |
| RIGHT_CLICK | x, y | Right click at the current position if x and y are not specified, otherwise right click at the specified position |
| DOUBLE_CLICK | x, y | Double click at the current position if x and y are not specified, otherwise double click at the specified position |
| DRAG_TO | x, y | Drag the cursor to the specified position with the left button pressed |
| SCROLL | dx, dy | Scroll the mouse wheel up or down |
| TYPING | text | Type the specified text |
| PRESS | key | Press the specified key and release it |
| KEY_DOWN | key | Press the specified key |
| KEY_UP | key | Release the specified key |
| HOTKEY | keys | Press the specified key combination |
| WAIT | - | Wait until the next action |
| FAIL | - | Decide the task cannot be performed |
| DONE | - | Decide the task is done |
| CALL_USER | - | Call the simulated user to fill the credentials when logging-in |

### B.5 ACTION SPACE

OS-MAP adopts the Computer_13 action space from OSWorld, covering all basic mouse and keyboard operations—such as mouse movement, various clicks, drags, key presses, and hotkeys. It also includes three meta-actions: WAIT, FAIL, and DONE, which allow the agent to express task progress or termination conditions.

To support human-in-the-loop collaboration, OS-MAP introduces a new action: CALL_USER, used when human input is required—for example, entering sensitive information like login credentials. This helps define the agent's permission boundary and enables more realistic human-agent cooperation. In OS-MAP benchmark, this action is only used during Google account login, where the agent yields control and a script autofills the credentials.

In total, OS-MAP defines 17 actions, summarized with their parameters in Table 4.

## C TASK CURATION DETAILS

All tasks in OS-MAP are designed according to a structured framework based on automation levels and hierarchies of user needs. Each task is implemented on the VM with a well-defined initial state and evaluation function, ensuring consistent and repeatable benchmarking. Task Curation was a collaborative effort by the authors, involving nine computer science students who jointly annotated and refined the tasks over approximately 600 hours of work.

Section C.1 outlines the standard six-stage pipeline for task creation, while Section C.2 provides a detailed walkthrough of how a representative Level-4 (L4) task was designed, iterated, and finalized from scratch. Section C.3 describes the process of filtering and re-annotating tasks imported from OSWorld Xie et al. (2024).

### C.1 PIPELINE DESCRIPTIONS

Based on the two-dimensional task organization framework described above, each task in OS-MAP is created by the co-authors following a standardized procedure:

Figure 12: Detailed Specification of the task goal in ToDo (not informed in the task instruction).

1. **Task selection.** We begin by identifying underrepresented scenes within the demand hierarchy. For each selected scene, we determine a representative application and outline a task concept aligned with that context.

2. **Exploration & specification.** Annotators study the target app or website using official documentation, demos, and hands-on interaction. They then define a concrete task objective, assign an appropriate difficulty level, and manually execute the task flow to verify feasibility. To prevent data contamination, task goals must not overlap with content from official materials; annotators are required to adapt or design new content accordingly.

3. **Instruction & configuration.** Annotators craft clear and concise task instructions and executable initialization configurations. Together, they control task difficulty—higher-level tasks omit details or include (human-recognizable) misleading cues, requiring agents to actively explore the environment for critical information.

4. **Reference state preparation.** The annotators manually complete the task to record a standard success state for the following evaluation process.

5. **Evaluation setup.** Evaluation involves comparing VM file or system states against predefined targets. Some tasks also require post-execution scripts or logic (postconfig) to expose the key status for assessment.

6. **Cross-validation.** Each task undergoes a rigorous review by two other annotators across several dimensions before inclusion: (1) task authenticity and representativeness, (2) clarity and unambiguity of instructions, (3) reproducibility, (4) correctness and (5) robustness of evaluation, (6) alignment with difficulty level, and (7) non-duplication.

### C.2 A REPRESENTATIVE EXAMPLE

We illustrate the creation of a representative `L4` task, from ideation to finalization:

1. **Task selection.** Upon reviewing the current task set, we found a gap in L4-level tasks within the office productivity domain—particularly tasks involving tool use, to-do management, and email communication. We thus defined a task prototype: write a to-do item that instructs the user to download multiple documents from a website, translate them using Google Translate, and send them as an email attachment to a colleague.

2. **Exploration & specification.** We selected the *How's Life* reports from the official OECD website as the document source. A to-do entry was added in a ToDo application, with a detailed task description specifying file names, save locations, and expected actions (see Figure 12).

3. **Instruction & configuration.** Through reverse engineering of the ToDo application, we identified the configuration file's location and edit protocol. Based on this, we created a config file and imported it during task initialization, so the to-do item loads automatically. Similarly, we reverse-engineered the Thunderbird email client, configuring its profile folder to pre-load an account and a draft email with a blank recipient field and no attachment.

4. **Reference state preparation.** We manually completed the task to obtain a reference success state—defined as the appearance of a new email in the recipient's local mail server directory. During testing, we observed long download times and limits on translation input and attachment size. To ensure feasibility, we reduced the requirement from translating six reports to just one.

5. **Evaluation setup.** The evaluation checks for textual equality between the expected and actual email file and is provided as part of the task package.

6. **Cross-validation.** The task was tested by two additional annotators to validate both procedure correctness and robustness—i.e., whether the task would still pass evaluation despite minor execution variations or small errors.

### C.3 FILTERING OF TASKS FROM OSWORLD

We reused and adapted the majority of tasks from OSWorld. After careful filtering and re-annotation, a total of 255 tasks were retained. The excluded tasks fall into three main categories:

- Redundant tasks within the same scenario – For example, the LibreOffice Calc tasks derived from the SheetCopilot dataset often involved templated spreadsheet operations (*e.g.*, statistical summaries and charting). Only 1–2 representative tasks were kept to avoid unnecessary duplication.

- Tasks lacking general relevance – These focused too heavily on application-specific UI details, such as fine-grained formatting combinations in office software (*e.g.*, font size, line spacing, paragraph alignment), rather than testing generalizable agent capabilities.

- Tasks marked as infeasible – These either had ambiguous descriptions or indirect, open-ended solutions that made evaluation problematic. For example, the task *"Could you please convert a PowerPoint presentation to video and play it with VLC?"* was removed. Retaining such tasks would conflict with our design principle of aligning higher-level tasks with feasible, goal-driven behavior, while rewriting them would introduce challenges in open-ended evaluation.

## D EXPERIMENT DETAILS

### D.1 MODEL BASELINES

We utilize the versions of `gpt-4o-2024-11-20` and `claude-3-7-sonnet-20250219` for results of `GPT-4o` and `Claude-3.7-Sonnet`, respectively, need to be noted that result could be changed from time since it is close-sourced. For all VLMs, we take the default hyper-parameters, *i.e.*, we set the temperature parameter to 1.0, and top_p to 0.9, and the maximum number of tokens for generation is set to 1500. We set the maximum number of interaction steps for `L1`, `L2`, `L3` and `L4` tasks to 15, 15, 30, and 50, respectively, which is sufficient to complete most tasks.

## E PROMPTS FOR AGENTS

Multi-modal computer use agent baseline involves complex prompt engineering, including system prompts, task instruction prompts, and step prompts. The following sections introduce these three types in detail and present representative examples of task instructions.

### E.1 SYSTEM PROMPT

The system prompt is the main part of prompt engineering in OS-MAP, including role description and observation space, action space, use cases, and format description with tips. The following will show the four parts of the system prompt. The complete system prompt is the splicing of the four parts.

**Role description and observation space**

```
You will act as an agent responsible for automating desktop
↪   computer tasks according to my instructions. You must possess
↪   strong knowledge of computer GUI operations and experience
↪   using common software applications.

For each task, I will provide you with an instruction that
↪   describes the task goal and may include additional hints. You
↪   will then enter an operation loop, where you fully take over
↪   the control of the computer, performing one action step at a
↪   time. At each step, you will receive the history of actions and
↪   the current screenshot as observations, and you need to output
↪   what action to perform next. The action will be executed, and
↪   the loop continues with a new screenshot.

Your output can include your reasoning--such as your observations,
↪   long-term planning, the objective of the current step, and the
↪   expected outcome. However, you must ALWAYS include a predicted
↪   ACTION and the action must conform to the action space
↪   described below. Your output must follow the specified FORMAT
↪   and include the correct `action_type` and required parameters.
```

**Action space**

```
ACTION_SPACE = [
    {
        "action_type": "MOVE_TO",
        "note": "move the cursor to the specified position",
        "parameters": {
            "x": {
                "type": float,
                "range": [0, X_MAX],
                "optional": False,
            },
            "y": {
                "type": float,
                "range": [0, Y_MAX],
                "optional": False,
            }
        }
    },

    ... more action definitions ...

    {
        "action_type": "TYPING",
        "note": "type the specified text",
        "parameters": {
            "text": {
                "type": str,
                "range": None,
                "optional": False,
            }
        }
    },

    ... more action definitions ...

    {
        "action_type": "CALL_USER",
        "note": "Call the user to fill in the Google account or
        ↪  password (the input box must be activated), one at a
        ↪  time, according to the parameter call_type.",
        "parameters": {
            "call_type": {
                "type": str,
                "range": ["email", "password"],
                "optional": False,
            }
        }
    },
    {
        "action_type": "WAIT",
        "note": "wait until the next action",
    },
    {
        "action_type": "FAIL",
        "note": "decide the task is failed or can not be
        ↪  performed",
    },
    {
        "action_type": "DONE",
        "note": "decide the task is done",
    }
]
```

**Use Cases**

```
Notes:
1. To reiterate, regardless of whether you include reasoning, your
↪  output MUST contain an action in the SPECIFIED FORMAT (a
↪  dictionary enclosed in triple backticks as shown in the
↪  examples below), and it must include a valid `action_type` and
↪  parameters as defined above.
2. For `MOUSE_MOVE`, you must specify the exact target `x` and `y`
↪  coordinates. The screen bounds are `X_MAX = 1920`, `Y_MAX =
↪  1080`. The coordinates must fall within [0, 1920] and [0, 1080].
↪  Example:
```{
  "action_type": "MOUSE_MOVE",
  "x": 1319,
  "y": 65
}```
3. For `[CLICK, RIGHT_CLICK, DOUBLE_CLICK, DRAG_TO]`, specifying `x`
↪  and `y` is optional. If omitted, the action defaults to the
↪  current cursor position (often used after `MOUSE_MOVE`).
↪  However, it is RECOMMENDED to specify the coordinates
↪  explicitly. Same format as `MOUSE_MOVE`:
```{
  "action_type": "CLICK",
  "x": 1319,
  "y": 65
}```
... more use cases ...
```

**Format descriptions with tips**

```
11. Other special actions are `[WAIT, FAIL, DONE]`. Use them when
↪  you think it's necessary to wait, when the task has failed, or
↪  when it has succeeded. Each `WAIT` pauses for ~2 seconds. Do not
↪  declare `FAIL` lightly without attempting reasonable actions and
↪  explorations, but if you still cannot get out of the
↪  predicament after trying for several steps, you can declare it.
↪  Only use `DONE` when you are certain the task is completed
↪  successfully. Do NOT use dictionary format or triple backticks.
↪  Just output like the BARE "DONE" without any other thought or
↪  formatting.
12. If the task is file editing, make sure it is saved successfully.
↪  If there is no clear description of the file name, save
↪  location, etc., use the default.
13. If there are clear step-level instructions, please follow them
↪  strictly. Otherwise, you can do whatever you want as long as
↪  the task is completed.
14. My computer password is `"password"`. You may use it freely
↪  whenever `sudo` access is required.

Please think step by step. Carefully observe the current screenshot
↪  and then output your reasoning (optional), your plan, the
↪  current action and expected results, and most importantly, the
↪  FORMATTED ACTION.
Do NOT ask questions. Do NOT attempt to interact with the user in
↪  any way other than via the `CALL_USER` action. You are fully
↪  responsible for controlling the computer. Do NOT output
↪  anything else.
```

### E.2 TASK INSTRUCTION PROMPTS

The task instruction is appended directly after the system prompt and is also loaded only once per task. It specifies the concrete objective the agent is expected to complete, in the following format:

```
You are asked to complete the following task: {{Instruction}}
```

### E.3 STEP PROMPT

The system prompt is loaded once at the beginning of each task to provide the agent with general instructions and behavioral priors. At every step during task execution, the agent also receives a step prompt in the following fixed format, where "History" stands for the previous three interaction history.

```
{{History}} {{Screenshot}} Given the screenshot below, what is the
↪   next step you will take to help complete the task?
```

### E.4 REPRESENTATIVE TASK INSTRUCTIONS

Task instructions are written in a human-friendly tone, clearly stating the objective and specifying any necessary details for evaluation. To enhance generalization and reduce overfitting to prompt patterns, we ensure diversity in language style and phrasing across tasks. Below, we present several representative examples.

**Rotate Wallpapers (`L4`)**

```
Download the Bing wallpaper for Italy from the latest 5 days in 4K
↪   resolution to ~/Pictures/wallpapers and name them 0.jpg
↪   (today's), 1.jpg, ..., add them to the wallpaper candidates,
↪   and set the today's one as the wallpaper. Next, configure a
↪   cron task to switch wallpapers in order at 00:00 every day. The
↪   required script change_wallpaper.sh is already provided on the
↪   desktop and can be used for cron tasks after only modifying the
↪   wallpaper directory.
```

**Zotero Citation (`L4`)**

```
I am writing my course paper and I need to cite a reference. I
↪   remember it in Zotero, but I can't remember which one it is.
↪   Please help me find this article and imitate the two IEEE
↪   formats above to complete the citation. Note that the font
↪   format must also be the same. Then download this article to the
↪   Documents/references folder and put it together with other
↪   cited papers.
```

**Meet Schedule (`L3`)**

```
Did you see the meeting time sent in the DATA group in Rocket.Chat?
↪   Add it to the event in Calendar, title it Team Meeting, and
↪   make sure you don't make mistakes with the date, time, and
↪   location.
```

# F    CASE ANALYSIS

This section provides the full context and failure analysis for several representative error cases referenced in the main Analysis section §4.3.

Figure 13 illustrates a severe hallucination, where the agent mistakenly identifies the current webpage as a Chrome browser interface and treats the top search bar as a search engine input.

Figures 14 and 15 show two `L3` tasks in which the agent fails to adapt when the straightforward method breaks down—for example, when the target element is missing from the screenshot.

In Figure 16, another `L3` task requires the agent to interact with a map embedded on the current page. However, the agent ignores this context and instead jumps to a global Google Maps search, bypassing the intended interaction.

Figures 17 and 18 depict two `L4` tasks characterized by long instructions and complex dependencies. In one case, the agent fails to properly decompose the instruction and proceeds with aimless exploration; in the other, it confuses the order of contextual navigation and concrete operations. These cases highlight the agent's significant shortcomings in handling the high-level reasoning and planning required at `L4`.

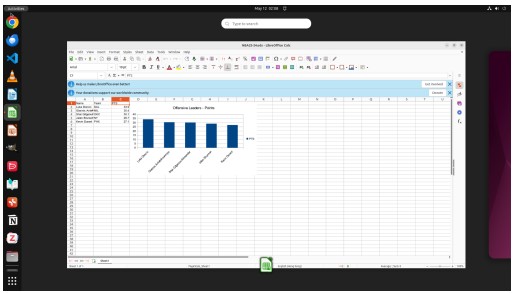
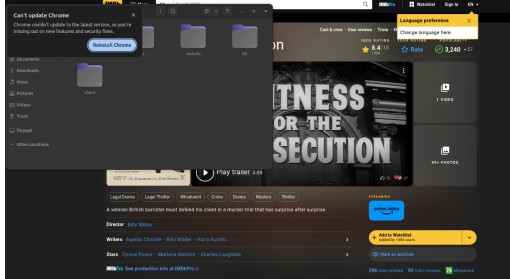

Figure 13: (`L2`) The agent identifies this page as Chrome and attempts to use the "search engine".

Figure 14: (`L3`) The agent does not know to use the keyboard shortcut to exit full-screen mode.

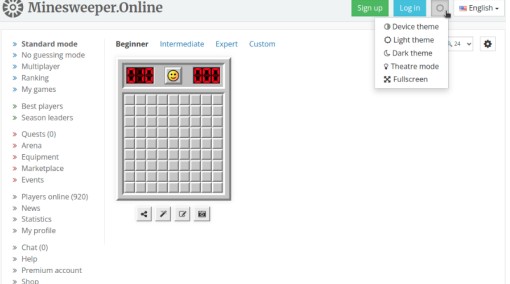
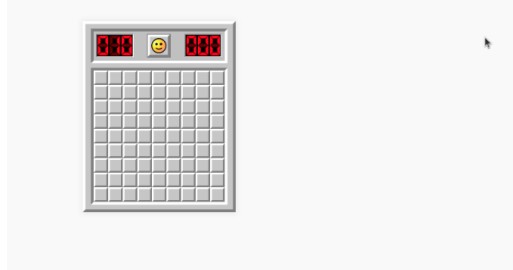

Figure 15: (`L3`) Task instruction: *Enter theater mode and resize the scale to 48.* However, the resize button is hidden in theater mode, and the agent does not know it should swap the execution order.

# G    LIMITATIONS AND FUTURE WORK

First, designing tasks that align precisely with automation levels and hierarchical needs often requires carefully controlled initial states and evaluation functions built through extensive reverse engineering, limiting the scalability of synthetic approaches. Moreover, the need for distribution and reproducibility prevents alignment with many real-world scenarios, which are often tightly coupled with user accounts, personalized content, or external effects, making them unsuitable as benchmark tasks.

Future work may explore scalable methods to generate fine-grained, controllable tasks that better cover the full range of user needs. Additionally, integrating environment-aware reward shaping could enable finer supervision and continual improvement for computer-using agents. We hope OS-MAP offers a solid foundation and actionable insights for advancing this direction.

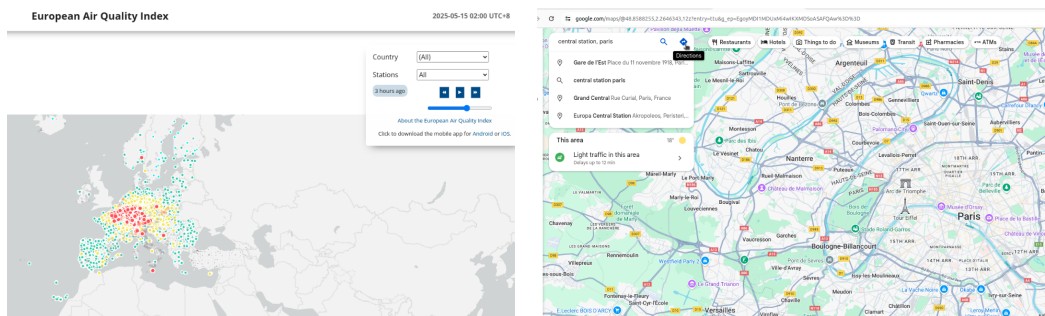

Figure 16: (L3) Task instruction: *Locate the MOST geographically central station in Paris on this map and jump to its location on Google Maps.* The agent simply ignores the current page (weather station map) and searches for subway stations in the center of Paris on Google Maps.

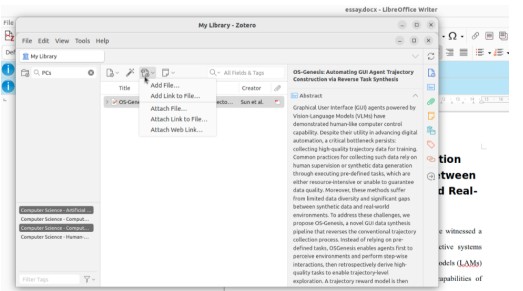

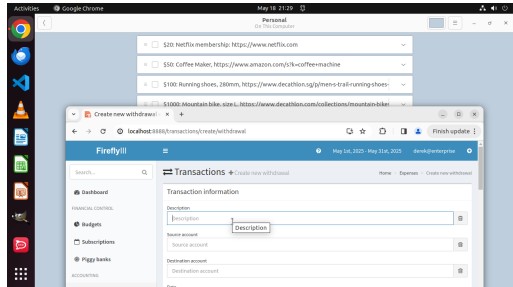

Figure 17: (L4) Task instruction: *I am writing my course paper and I need to cite a reference. I remember it in Zotero, but I can't remember which one it is. Please help me find this article and imitate the two IEEE formats above to complete the citation. Note that the font format must also be the same. Then download this article to the Documents/references folder and put it together with other cited papers.* The agent did not break down the task into subtasks and clicked aimlessly on the Zotero interface.

Figure 18: (L4) Task instruction: *I plan to use the money in my wallet to buy something to reward myself. Please check how much money is in my wallet account, and then buy the item of the corresponding amount in the todo list. Please choose the appropriate size and add it to the shopping cart. Then go back to the firefly and add a corresponding expense transaction, named the item name on the todo list.* The agent does not check the wallet balance or place an order, but tries to add a transaction record first.

## H    REPRODUCIBILITY STATEMENT

We provide an anonymous downloadable source code at https://anonymous.4open.science/r/OSMap-C2F5/. The deployment process of OS-MAP is detailed in the README.md of the code repository, while the experimental settings for running evaluations on OS-MAP are described in Section 4.

