# OpenReview forum: "OS-MAP: How Far Can Computer-Using Agents Go in Breadth and Depth?"
_ICLR.cc/2026/Conference — Submitted to ICLR 2026_

### Official Review · Reviewer_XYdi · 2025-10-29

**Soundness:** 3
**Presentation:** 3
**Contribution:** 3
**Rating:** 4
**Confidence:** 4

**Summary:**

This paper introduces OS-MAP, a new benchmark for evaluating computer-using agents. The authors argue that existing benchmarks fail to capture the true diversity of real-world tasks and treat them as a "flat" collection, which prevents the measuring of critical agent capabilities like autonomy and generalization, or aligning tasks with actual user demands.

The primary contribution is an OS agent evaluation benchmark consisting of 416 tasks across 15 applications, and the novel two-dimensional framework used to organize it. This framework evaluates agents along two axes: Automation Level and Generalization Scope.

In experiments section, the authors evaluate a range of agents and showing all current agents perform poorly, achieving only an 11.5% overall success rate and near-zero performance on higher-level automation tasks (L3/L4).

**Strengths:**

This paper makes a valuable contribution to existing agent benchmarks, such as OSWorld, by enriching the set of tasks and carefully reorganizing and refining the benchmark categories. These additions broaden the benchmark’s coverage and help establish a more comprehensive evaluation framework for agent capabilities.

The experimental evaluation is solid and thorough. The authors provide both code and data, demonstrating strong reproducibility and completeness of the work.

Overall, the paper’s core ideas are clearly articulated, and the presentation is well structured, making the contributions easy to understand.

**Weaknesses:**

While the paper presents meaningful contributions, there are a few areas where it could be strengthened.

First, Appendix C contains important details about task curation, but these details currently sit outside the main narrative. Since task design is a core component of the benchmark, I would encourage the authors to move at least some of this content into the main paper to better motivate and justify the task set.

The proposed difficulty-level formulation also feels only partially novel. Similar ideas have been discussed in prior work, such as VideoWebArena (2024), which also proposes difficulty-based categorization.

Regarding Figure 1, the mapping of “proactivity” and “general” as higher-value areas is not entirely convincing. Basic execution tasks are equally important for evaluating agent capabilities. These axes might be better positioned as indicating “more challenging regions” rather than inherently “more valuable.”

I also found the proposed automation-level taxonomy unconvincing. Although I see the motivation, the categories don’t fully capture the range of challenges GUI agents face. Beyond task ambiguity, issues such as long-horizon execution, UI complexity, tool availability also significantly affect difficulty. For example, in Figure 3, I would not consider “download and use today’s Bing wallpaper” and “download and rotate the Bing wallpaper every day” to represent distinct automation levels. The distinction feels minor. The authors may want to reconsider this taxonomy, especially rethinking about adapting from SAE’s driving automation taxonomy as OS agent and driving are naturally different.

**Questions:**

see above

---

> ### Author Response · Authors · 2025-11-21
> **Response to Reviewer XYdi**
>
> Thanks for your insightful comments. We decompose your review text into the following issues:
>
> > **1. Appendix C contains important details about task curation, but these details currently sit outside the main narrative.**
>
> We sincerely appreciate your valuable suggestion. Appendix C briefly introduces OSWorld’s task creation pipeline, a representative task creation process, and task filtering process.
>
> Due to page limitations, these details are not included in the main text. In the revised version, we will adopt your suggestion: relocate non-core content to the appendix to free up space and integrate this important content into the main body.
>
> > **2. Difficulty-level formulation also feels only partially novel.**
>
> You are correct that similar difficulty-based grading designs have been discussed in works such as [VideoWebArena](https://arxiv.org/abs/2401.13649v2) and [AndroidAgentArena](https://arxiv.org/abs/2501.01149v2).
>
> However, OS-Map’s core contribution lies in its distinct grading logic: unlike aforementioned works that simply divide difficulty ranges by **execution steps**, we decompose the core capabilities required for Computer Use Agents to complete tasks and define grading criteria based on the **progressive advancement of these capabilities**.
>
> This approach is more specific, clear, and informative than step-based division. We aim for OS-Map’s evaluation results to support detailed assessments of current agents’ existing/lacking capabilities and targeted improvements in future work—whereas step-based difficulty hierarchies only yield vague conclusions (e.g., "insufficient long-range planning capabilities").
>
> > **3. The mapping of “proactivity” and “general” as higher-value areas**
>
> We sincerely appreciate your valuable suggestion and fully agree with your perspective. In the revised version, we will revise "higher-value areas" in Figure 1 to "more challenging regions."
>
> > **4. About the proposed automation-level taxonomy**
>
> We sincerely appreciate you pointing out this concern and discuss it in detail below:
>
> - First, we endorse your perspective on task difficulty and have included corresponding tasks in OS-Map.
>     - L4 tasks emphasize long-horizon execution (>15 steps) and tool availability: e.g., task [`l4_course_parse_tables`](https://anonymous.4open.science/r/OSMap-C2F5/evaluation_examples/examples/multi_apps/l4_course_parse_tables.json) provides a table-parsing script (no instructions) for correct use in extracting course website tables.
>     - UI complexity (hard to quantify) is not controlled; it is implicitly covered/averaged via multi-task S-axis scenario variation. We focus on task completion, not overly complex UI understanding.
> - Second, "download and rotate the Bing wallpaper every day" is not a daily repetition of "download and use today’s Bing wallpaper". It requires downloading past wallpapers, saving to specified path, renaming, and writing crontab scheduled scripts. Longer workflow, cross-app interaction, and context switching are L4’s incremental complexity vs. L3 (see task [`l4_bing_wallpaper`](https://anonymous.4open.science/r/OSMap-C2F5/evaluation_examples/examples/multi_apps/l4_bing_wallpaper.json)).
> - Finally, OS-Map draws on SAE driving framework’s **user-system responsibility-based grading**, but detailed criteria are not replicable due to the significant differences. Our standards fully consider common computer use scenarios and core capabilities.

---

> > ### Comment · Reviewer_XYdi · 2025-11-27
> >
> > Thanks author for the explanation. My concerns have been cleared and I have raised the score

---

> > > ### Author Response · Authors · 2025-11-28
> > > **Response to Reviewer XYdi**
> > >
> > > Dear reviewer,
> > >
> > > We are delighted to hear that our response addressed your concerns! We will incorporate the additional details and suggestions during the discussion period into the revised paper.
> > >
> > > Thank you once again for your insightful review and support.

---

### Official Review · Reviewer_bYpX · 2025-10-31

**Soundness:** 4
**Presentation:** 4
**Contribution:** 4
**Rating:** 10
**Confidence:** 3

**Summary:**

This paper introduces OS-MAP, a benchmark with 416 real-world computer-use tasks on 15 Ubuntu applications. Different from existing benchmarks like OSWorld, this benchmark's captures two orthogonal dimensions: automation level and generalization scope. The paper then performs a detailed set of experiments on a set of models and agents and found them to significantly underperform human.

**Strengths:**

- This paper is extremely well written
- It provides a very comprehensive benchmark, covering a good range of variety on multiple dimensions
- The benchmark enables detailed analysis of agent performance and failure cases
- It conducts a very thorough evaluation of popular models and agents and found interesting results.

**Weaknesses:**

None. I really like this paper

**Questions:**

None. It's very clearly written.

---

> ### Author Response · Authors · 2025-11-21
> **Response to Reviewer bYpX**
>
> We sincerely appreciate your positive feedback and full recognition of our work. We are delighted that the clarity of writing, comprehensiveness of OS-MAP, and depth of evaluations have resonated with you.
>
> If you have any new questions or discussions during the review period, please feel free to raise them. We will respond carefully and address your concerns thoroughly.
>
> Thank you again for your valuable support and kind recommendation; we will continue to revise the manuscript to ensure it meets the standards.

---

### Official Review · Reviewer_VFWi · 2025-10-31

**Soundness:** 3
**Presentation:** 3
**Contribution:** 3
**Rating:** 4
**Confidence:** 3

**Summary:**

This paper proposes a systematic benchmark for evaluating computer-operated intelligent agents—OS-MAP—aimed at comprehensively characterizing the automation capabilities and generalization levels of agents in real desktop environments. The authors construct a high-fidelity interactive environment based on a virtual machine, incorporating 416 tasks covering 15 common application scenarios, and propose a two-dimensional evaluation framework: five levels of automation and three levels of generalization. Under this benchmark, the authors systematically evaluate several mainstream models, and the results show that the overall success rate of the most powerful agents is currently far lower than that of human agents. The paper further reveals the key bottlenecks of agents through hierarchical failure analysis, providing a clear development path and a scalable evaluation framework for future research.

**Strengths:**

1. **Unified capability framework.** The paper proposes an original two-axis evaluation scheme—automation level × generalization scope —that provides a structured view of agent competence and offers a clear roadmap for future development.

2. **High-fidelity and reproducible benchmark.** OS-MAP builds on a virtualized desktop environment that spans realistic domains such as office work, study, and system management. The setup ensures controlled reproducibility while remaining extensible to new applications.

3. **Comprehensive experimentation and analysis.** The study benchmarks both proprietary and open-source models, presenting quantitative comparisons and qualitative failure analyses. The results convincingly highlight the main obstacles faced by current CUAs in higher-level (L3/L4) tasks, such as long-horizon planning and environmental adaptation.

**Weaknesses:**

1. **Limited evaluation metrics.** The benchmark currently uses binary task-success scores (0/1), ignoring partial progress, efficiency, or robustness. Incorporating richer quantitative indicators—e.g., partial completion ratio, average steps per success, or recovery rate—would provide a more nuanced evaluation.

2. **High manual cost of task construction.** Although OS-MAP follows a standardized six-stage pipeline, creating and validating new tasks still requires substantial human effort and reverse engineering. This limits scalability if the benchmark is to expand beyond 400 + tasks.

3. **Single-round task execution.** Current tasks evaluate one-shot command execution rather than continuous interaction or lifelong learning.As a result, the benchmark cannot yet assess persistent collaboration or long-term adaptation capabilities.

4. **Bias toward GUI-level operation tasks.** Most tasks emphasize interface manipulation rather than semantic-level reasoning or information synthesis. Extending OS-MAP to include tasks that require multi-step reasoning or cross-application content understanding would strengthen its coverage.

**Questions:**

1. **L5 proactive tasks.** The paper defines a “Proactive Companion” (L5) level but does not include any concrete L5 tasks or evaluation criteria. Do the authors plan to incorporate proactive or context-aware tasks in future versions of OS-MAP?

1. **Task-generation automation.** Given the manual workload, have the authors explored using LLMs or GUI-recording tools to automatically generate task descriptions and initialization scripts?

1. **Cross-platform generalization.** All current experiments are conducted on Ubuntu. Have the authors evaluated portability to other operating systems? A cross-platform benchmark would improve external validity and demonstrate the robustness of the environment design.

---

> ### Author Response · Authors · 2025-11-21
> **Response to Reviewer VFWi**
>
> Thank you for your detailed review. We sincerely appreciate your response and recognition and will address your concerns below.
>
> > **W1: Limited evaluation metrics**
>
> We acknowledge our current evaluation uses only binary (0/1) task completion outcomes, without dense process-oriented metrics. This stems from Computer Use tasks’ inherent traits: open dynamic environments and long, non-unique implementation paths—making intermediate evaluators hard to design and model-based assessments inaccurate due to system/software complexity.
>
> To ensure framework consistency and benchmark robustness, we adopt a purely rule-based evaluation (following prior works), trading some flexibility. We fully endorse your insight: process-oriented metrics would enable multi-dimensional agent evaluation. In future iterations, we will add key process metrics, such as human reference path average step length and qualitative execution error information.
>
> > **W2: High manual cost of task construction**
>
> We acknowledge current task creation and validation rely heavily on manual effort (~3 hours per task, covering design, creation and verification). Given Computer Use tasks’ complexity, our original goal is high-quality, generalizable data. We attempted automated task collection via models (e.g., OS-Genesis) but prioritized benchmark quality and test accuracy, ultimately adopting manual annotation with cross-validation.
>
> > **W3: Single-round task execution**
>
> We acknowledge our tasks primarily focus on one-shot command execution, and elaborate on two aspects:
> 1. For L1–L4 tasks, online and continuous learning can be achievable without framework modifications—only by adding process rewards for each task. The open environment encourages future reinforcement learning environment setups to enhance agent capabilities via dynamic exploration and trial-and-error.
> 2. L5 tasks (envisioned for continuous interaction and lifelong learning) require a distinct evaluation framework, so they are presented as an outlook rather than concrete implementation. We hope future research will focus more on L5 capabilities.
>
> > **W4: Bias toward GUI-level operation tasks**
>
> We respectfully disagree with this perspective.
> - Only L1 tasks in OS-Map are purely GUI-level operation tasks.
> - L2 tasks require the integration of semantic understanding and reasoning.
> - L3 tasks further demand adaptability, implicit information acquisition, error detection, and recovery capabilities.
> - L4 tasks emphasize high-level sub-task decomposition, cross-application collaboration, and long-context management.
>
> As the automation level advances, tasks impose increasing demands on multi-dimensional capabilities such as semantic comprehension, information extraction, long-range planning, multi-step reasoning, and error recovery.
>
> - For example, the L4 tasks in Figure 3: "Download & rotate Bing wallpaper every day" requires the synthesis of the aforementioned capabilities, and pure GUI operation skills are far from sufficient.
> - Quantitatively, Table 1 shows that **154 tasks (37.7%)** involve cross-application interactions across two or more apps, while all **42** L4 tasks entail scenarios with **three or more applications**. This constitutes one of our key contributions.

---

> > ### Author Response · Authors · 2025-11-21
> > **Response to Reviewer VFWi (Cont.)**
> >
> > > **Q1: L5 proactive tasks**
> >
> > We acknowledge L5 tasks are not yet included, primarily to maintain benchmark evaluation logic consistency. L1–L4 share a unified responsive evaluation environment, while L5 requires a customized setup (predefined user needs, agent ground truth support) that differs from L1–L4’s environmental and state evaluation logic.
> >
> > For completeness, future work will attempt a dedicated L5 evaluation environment, striving to align with OS-Map’s core design principles. We will also continue following advancements in proactive GUI agent scope exploration, a field actively studied currently.
> >
> > > **Q2: Task-generation automation**
> > As noted in W2, substantial human effort is OS-Map’s primary limitation.
> >
> > We conclude that LLMs assist in task scope/proposal generation, but application reverse engineering and repeated correctness validation (indispensable for executable benchmarks) exceed their capability—LLMs lack reliable specific detail knowledge.
> >
> > We thus use a hybrid approach: LLMs for task proposals, human annotators for validity/suitability verification to ensure quality.
> >
> > In practice, our self-developed lightweight GUI recording tool replays human operations (precise bounding boxes/action types) but does not assist in task design.
> >
> > > **Q3: Cross-platform generalization**
> >
> > Your suggestion is valuable but not our short-term top priority:
> >
> > 1. OS-Map requires open-permission system-level tools and avoids commercial, strict-permission, or account-required software. Thus, we use Ubuntu OS and open-source daily software alternatives (e.g., LibreOffice instead of Microsoft Office).
> > 2. Its human-like I/O design ensures good universality—agents need no underlying system adaptation. Per human experience, proficiency in one OS enables quick adaptation to others and their apps. While system-level operation differences exist, they are not our current focus; our primary goal is a generalized, feature-rich testing ground.
> >
> > Thank you for your suggestion. We will consider migrating to other OSes in the future to address inherent system-level operation differences and enhance cross-platform universality and robustness.

---

> > ### Comment · Reviewer_VFWi · 2025-11-28
> >
> > Thank you for the detailed rebuttal and clarifications.
> >
> > * Addressed concerns. The authors have convincingly explained the design choice of using binary end-state evaluation and its alignment with prior work, and have outlined reasonable plans to add process-oriented metrics in future versions. The discussion on manual task construction (including time cost, attempts at automated pipelines, and the use of GUI recording plus LLM-assisted proposal) also addresses my concern that this was an ad-hoc choice. The answers regarding L5 tasks, task-generation automation, and cross-platform support clearly delineate the current scope and plausible future extensions.
> >
> > * Remaining concerns. I still view the current benchmark as primarily focused on single-round execution, without directly evaluating persistent or multi-round interaction. In addition, although the clarification on higher-level tasks mitigates my original worry about a “GUI-only” focus, I believe there is still room to incorporate more explicitly semantic-heavy tasks in future iterations.
> >
> > These clarifications are helpful, but the remaining limitations still affect my overall view of the current version.

---

> ### Author Response · Authors · 2025-12-03
> **Response to Reviewer VFWi (Cont.)**
>
> Dear Reviewer,
>
> Regarding your remaining concerns, we would like to offer a brief clarification to further align our perspectives:
>
> > On "Single-round execution" vs. Multi-round Interaction:
>
> We understand your concern regarding "persistent" interaction. We would like to clarify the distinction between "Episodic Long-horizon Interaction" (our current focus) and "Lifelong Interaction".
>
> Current State: While OS-MAP tasks are episodic (resetting state between tasks), the interaction within a task is highly multi-round and long-horizon. As noted in Table 1, tasks average 11.4 steps, with L4 tasks often requiring 50+ steps of continuous perception, reasoning, and action loops. This goes far beyond "one-shot command execution."
>
> Value: Our results show that current SOTA agents (success rate ~11.5%) struggle significantly even with these episodic long-horizon interactions. We believe solving these distinct, complex episodes is a necessary prerequisite before agents can effectively handle persistent, cross-session collaboration (L5). OS-MAP serves as the stress test for this critical foundational capability.
>
> > On "Semantic-heavy tasks"
>
> We agree with your insight that there is room for tasks requiring even deeper semantic synthesis.
>
> Current Semantic Depth: We aimed to balance "GUI Grounding" and "Semantic Reasoning." Many L3/L4 tasks in OS-MAP do require processing dense information (e.g., reading a PDF report to extract specific data before emailing), rather than just mechanical clicking.
>
> Future Scope: We acknowledge that tasks focusing predominantly on complex reasoning (e.g., analyzing financial trends across multiple spreadsheets) are a valuable direction. We will explicitly discuss this limitation in the final version and highlight it as a roadmap for future extensions, potentially integrating computer-using with general, inference-heavy benchmarks like GAIA[1] to enrich the "Depth" dimension of our matrix.
>
> Thanks again for your comments that help us to improve the quality of this work!
>
> ### Reference
>
> [1] GAIA: a benchmark for General AI Assistants https://arxiv.org/abs/2311.12983

---

### Official Review · Reviewer_vHki · 2025-11-01

**Soundness:** 3
**Presentation:** 3
**Contribution:** 2
**Rating:** 4
**Confidence:** 3

**Summary:**

This paper introduces OS-MAP, a new benchmark designed to evaluate computer-using agents along two principal dimensions: breadth (generalization scope across user demands) and depth (automation level). The authors argue that existing benchmarks treat tasks as a flat collection, failing to capture the heterogeneity of real-world computer use and hindering progress towards practical deployment.

**Strengths:**

- The core contribution—the two-dimensional evaluation matrix of Automation Level vs. Generalization Scope—is a step forward for agent evaluation. It provides a structured, principled way to move beyond simple aggregate success rates.

- The "Generalization Scope" dimension is based on a well-researched user demand hierarchy, adapting mobile and desktop usage statistics. This demand-driven approach to task curation ensures that the benchmark's relevance is tied to practical utility, a crucial aspect often overlooked by benchmarks organized solely around applications or synthetic task templates.

-The paper evaluates a diverse set of modern agents, providing a comprehensive snapshot of the current state-of-the-art.

- The authors are releasing the benchmark, environment, and baselines. I took a look at the code and think it is valuable for commnuaity.

**Weaknesses:**

- The Conflation of Task Complexity and Agent Autonomy in the "L-Levels" is some what misleading. The framework's "Automation Levels" are inspired by the SAE levels for driving, but there's a crucial difference. In driving, the core task ("go from point A to point B safely") is constant. The levels define the division of labor and the operational design domain. In OS-MAP, the tasks themselves become fundamentally more complex at higher L-levels. L1 involves atomic actions, while L4 involves multi-step, multi-app workflows. This conflates two different things:

  - Task Complexity: The inherent difficulty, length, and abstractness of the user's goal.

  - Agent Autonomy: The agent's ability to operate without human intervention or guidance.

- The "Ladder" Fallacy: Are Automation Levels Truly Sequential?
The framework implies a progression: an agent must master execution (L1) before planning (L2), and planning before adaptation (L3). This "ladder" model doesn't capture the reality that these are often parallel, and sometimes orthogonal, skills.

- I do think the benefit of this benchmark but I does not agree on the eval framework.

**Questions:**

1. Could you elaborate on the process for assigning tasks to the automation levels L1-L4? Was there a formal rubric, and did you measure inter-annotator agreement to ensure consistency in the labeling? For example, the "Rotate Wallpapers" task is labeled L4, which involves downloading, setting a wallpaper, and configuring a cron job. What specific aspects make this L4 (Global Conductor) rather than a complex L3 (Adaptive Agent) task?

2. How do you envision quantitatively measuring an agent's performance on the S-axis (Generalization Scope)? Could you propose a metric based on the OS-MAP results to classify an agent as S1, S2, or S3?

---

> ### Author Response · Authors · 2025-11-21
> **Response to Reviewer vHki**
>
> Thank you for your detailed review. We sincerely appreciate your response and recognition and will address your concerns below.
>
> > **W1: The Conflation of Task Complexity and Agent Autonomy in the "L-Levels".**
>
> We argue Task Complexity and Agent Autonomy are interdependent (in autonomous driving and Computer Use scenarios) and ultimately denote this integrated dimension via Agent Autonomy.
> 1. The two scenarios share analogous core requirements:: Computer Use relies on abstract state transitions operationalized via human-aligned GUI steps; driving (Point A→B) needs concretization via real-world factors (distance, time, traffic, vehicle status).
> 2. Agent capabilities (autonomy) are defined solely by independent task complexity—simple tasks indicate specific capabilities, complex tasks reflect breadth/depth; autonomy without task complexity is meaningless.
> 3. Autonomy grading is determined by **user-agent responsibility division**, consistent with SAE principles. Both driving (location transitions) and Computer Use (task completion) are high-level goals, with technical implementations as specific means. Responsibility division frames evaluation of an agent’s "certain level of intelligence."
>
> > **W2: Are Automation Levels Truly Sequential?**
>
> OS-Map implicitly assumes progressive automation level advancement. However, as you astutely noted, these capabilities indeed have parallel or orthogonal relationships rather than being purely hierarchical. We simplified this complexity for two key reasons:
> 1. For Computer Use scenarios, a stepwise capability framework enables abstraction of an agent’s core competencies, avoiding overemphasis on scenario-specific details.
> 2. Our primary focus is agents’ ability to autonomously complete tasks of varying complexity. Thus, the core capability at each level not only raises the upper limit of task complexity but also **encompasses all lower-level** capabilities to ensure practicality. For instance, an agent with strong high-level orchestration but weak planning skills would struggle to independently complete complex tasks in reality.
>
> > **W3: Evaluation Framework Persuasiveness**
>
> We sincerely appreciate your recognition of OS-MAP’s contributions. As noted earlier, we acknowledge the current evaluation system is somewhat idealized, but the overall framework design offers a universal, extensible approach to GUI Agent evaluation—focusing on core essence while appropriately abstracting minor details.
>
> As other reviewers also suggested, we will explore additional evaluation paradigms and refine diverse skill datasets in future work to make OS-MAP more comprehensive and practical.
>
> > **Q1: More Detailed Grading Guidelines for L1–L4**
> - **L1:** Executes pre-decomposed atomic operations (e.g., clicking, typing), requiring basic GUI semantic understanding and grounding.
> - **L2:** Plans and executes multi-step operations (<15 steps), demanding enhanced Computer Use context comprehension and practical experience for simple planning.
> - **L3:** Extends L2 with long-tail scenarios (implicit context, unexpected changes, errors), requiring strong semantic comprehension and adaptability (responding to changes, exploring alternatives, error recovery). Annotators were instructed to identify or manually construct natural non-expected scenarios.
> - **L4:** Builds on L3, emphasizing task decomposition, cross-application (≥3) alternation, and long-context management (>15 steps). It also encompasses L3 capabilities (e.g., ad-immune Bing Wallpaper website localization). The "Rotate Wallpapers" task (sequential sub-goals, 3 apps, long workflow) is a typical example.
>
> All task proposals were generated by annotators following consistent capability standards and annotation rules with cross-validation to minimize subjective interference. Final task inclusion and grading were determined by the first author via comprehensive evaluation, to ensure consistency.
>
> > **Q2: Quantitative Measure of Generalization Scope**
>
>  Following the task scenario classification in Figure 4, we categorized instances into a three-level hierarchy. Beyond the overall average score, we also calculated subclass scores within this hierarchy:
>
> - **S1:** Completes 50% of first-level subcategories under any one second-level category (within a third-level category).
> - **S2:** Completes 50% of first-level subcategories across all second-level categories under any one third-level category.
> - **S3:** Completes 50% of first-level subcategories across all second-level categories under more than three third-level categories.
>
> In practice, agents’ execution capabilities were lower than expected: most tasks remained unachievable, and no model met the S1 criteria. Thus, we did not report S-axis evaluation results.
>
> We appreciate your insightful question. In future work, we will expand L-axis and S-axis task instances, balance their distribution across levels, and promptly update capability evaluation results as agent capabilities advance.

---

> > ### Comment · Reviewer_vHki · 2025-11-26
> >
> > Thank you to the authors for the detailed and thoughtful rebuttal.
> > After reviewing the responses, I find that several concerns remain insufficiently addressed:
> >
> > - Conflation of Autonomy and Task Complexity
> >
> > While the authors argue these dimensions are “interdependent,” this does not fully resolve the conceptual entanglement. The current L-level definitions continue to couple autonomy with an expansion of task complexity, making it difficult to disentangle whether performance differences arise from improved planning/agency or merely the agent’s ability to handle longer or more complicated workflows. This limits the evaluative interpretability of the hierarchy.
> >
> > - Sequential Nature of the Levels
> >
> > The rebuttal acknowledges that the hierarchy is idealized and simplifies parallel skill dimensions, but this simplification is precisely the issue: it obscures important distinctions across agent capabilities. The framework still implies a ladder-like progression that may not hold in practice, and the justification presented does not fully alleviate this conceptual rigidity.
> >
> >
> > Overall, while the benchmark itself appears valuable and the release of the environment and baselines is a contribution, the conceptual issues with the evaluation framework remain unresolved in my view.
> >
> > **Final Recommendation**
> >
> > I appreciate the authors’ efforts and clarifications, but the rebuttal does not sufficiently address the core conceptual concerns regarding the L-axis structure and the interpretability of the evaluation framework. I am therefore maintaining my original score. Confidence remains unchanged.

---

> > > ### Comment · Reviewer_vHki · 2025-11-27
> > >
> > > I appreciate the authors’ rebuttal. However, the responses do not sufficiently resolve the conceptual concerns raised in my original review.
> > >
> > > - Conflation of Autonomy and Task Complexity
> > >
> > > The rebuttal reiterates that autonomy and complexity are “interdependent,” yet this explanation does not address the core issue: the current L-scale makes it impossible to interpret whether an agent’s failure is due to insufficient autonomy or simply increased task difficulty. The framework still binds the two notions together in a way that reduces diagnostic value. Clarifying that the levels are “integrated” does not resolve the conceptual ambiguity this introduces for evaluation and comparison.
> > >
> > > - Sequential vS. Parallel Capabilities
> > >
> > > While the authors acknowledge the hierarchy is idealized,  the L-levels continue to impose a ladder-like progression that does not reflect how agent competencies typically develop. Many abilities at L3 or L4 (e.g., error recovery) do not logically presuppose proficiency at earlier levels, and the framework does not accommodate such non-sequential capability distributions.
> > >
> > > Given these unresolved conceptual issues, the rebuttal does not materially change my assessment of the evaluation.

---

> > > > ### Author Response · Authors · 2025-12-03
> > > > **Response to Reviewer vHki (Cont.)**
> > > >
> > > > Thank you for your continued engagement. We genuinely appreciate your critical thinking regarding the theoretical part of our evaluation framework. We have carefully reflected on your comments about the "Conflation of Autonomy/Complexity" and the "Sequential Nature."
> > > >
> > > > We would like to offer a further clarification on how we intend to address these points without altering the core experimental results, but rather by enhancing the interpretation and analysis.
> > > >
> > > > > On the Nature of Taxonomy
> > > >
> > > > We agree with that **no taxonomy for general-purpose agents is perfect**, and any attempt to categorize complex behaviors involves inherent trade-offs.
> > > >
> > > > We acknowledge that our "L-levels" are an idealized abstraction designed to facilitate standardized benchmarking, rather than a flawless biological model of intelligence development. We will explicitly discuss this "debatable nature" in the Limitations. We will clarify that our taxonomy is one effective lens for organizing tasks, chosen for its alignment with the SAE operational model, but it is not the only possible lens.
> > > >
> > > > > Clarifying "Conflation" vs. "Correlation."
> > > >
> > > > We maintain that in a benchmark setting, Agent Autonomy is unobservable without Task Complexity. OS-Map is not "conflating" them conceptually, but rather linking them operationally for evaluation purposes. An agent's autonomy can only be verified by its success in complex environments (e.g., L4 Orchestration ).
> > > >
> > > > We will refine the text to ensure readers understand that L-levels serve primarily as "Task Difficulty Indicators" that proxy for autonomy, reducing the ambiguity you rightly pointed out.
> > > >
> > > > > Addressing "Parallel Capabilities"
> > > >
> > > > We accept your valid point that skills (like error recovery vs. planning) are often orthogonal. While we will retain the L-levels as the primary organization structure for consistency, we will further expand exisiting analysis and annotate more tasks (which require time and resources) and incorporate a "Capability-wise Analysis" (e.g., separating Perception, Grounding, and Reasoning) in the analysis section.
> > > >
> > > > At last, we believe OS-MAP’s value lies in its robust infrastructure and data (416 human-validated tasks, 15 apps, executable environment), which serves as a critical resource for the community. While the evaluation **taxonomy may remain a subject of open academic debate**, we hope the practical utility of the benchmark and our commitment to providing nuanced capability analysis (beyond just L-levels) will merit your support.
> > > >
> > > > Thanks again for your comments!

---

### Author Response · Authors · 2025-12-03
**General Response and Summary of Rebuttal Updates**

Dear Area Chair,

We sincerely appreciate the additional efforts invested in handling the submissions under the current circumstances. To assist your reassessment after the review rollback, we provide a summary of the author–reviewer discussions. We confirm that we have engaged in the discussion strictly in accordance with the ICLR Code of Conduct.

First, we are encouraged by strong positive feedback from the reviewer `bYpX`. Then, during the discussion, we actively engaged with all reviewers to address their questions. As a result, Reviewer `XYdi` explicitly raised the rating after our clarifications.

| Reviewer | Initial Score | Main Concerns / Request                                                      | Our Response                                                                                                                                                  | Status / Updated Score |
|----------|---------------|------------------------------------------------------------------------------|---------------------------------------------------------------------------------------------------------------------------------------------------------------|------------------------|
| `bYpX`     | 10            | Identified no significant weaknesses; strongly recommended.                  | Will further revise the quality of manuscript and evaluation framework                                                                                                                                                           | 10 (No Reply)          |
| `XYdi`     | 4             | Task taxonomy logic; appendix organization; value of "proactivity".          | Clarified L-level taxonomy (vs. step-based); agreed to move task curation details to main text in the final version.                                          | 6 (Score Updated)      |
| `vHki`     | 4             | Conceptual definition of "Automation Levels" (Autonomy vs. Task Complexity). | Explained SAE alignment: Autonomy and complexity are interdependent in realistic evaluations.                                                                 | 4 (Discussion Ongoing)         |
| `VFWi`     | 4             | Limited metrics (binary only); manual cost; single-round execution focus.    | Justified robust rule-based eval; explained hybrid task creation pipeline (human involvement are required to ensure benchmark quality); clarified L5 outlook. | 4 (Discussion Onging)         |

Summary of Discussion Outcomes:

- **Consensus on Value**: It is worth noting that the core value and quality of OS-MAP received recognition. **No reviewer questioned the validity, quality, scale, or necessity of the benchmark**. Reviewer `bYpX` (10) and `XYdi` (4->6) highlighted its comprehensive scope.

- **Progress with Reviewer VFWi**: Reviewer VFWi acknowledged that our initial rebuttal *convincingly explained the design choice... and addressed my concern* . Regarding the further comment on 'single-round execution,' we provided more clarification explaining that OS-MAP tasks, while episodic, are long-horizon (avg 11.4 steps, up to 50+), requiring continuous perception-action loops far beyond simple "one-shot" commands. We believe this clarification bridges the gap to `VFWi`'s expectations.

- **Clarifications on Concepts**: regarding Reviewer vHki, the ongoing discussion focuses on a conceptual difference in defining "Automation Levels." While the reviewer perceives a conflation between autonomy and complexity, we have clarified that our design intentionally aligns with SAE levels to reflect the interdependence of these factors in realistic agent evaluation. We believe this is **a difference in perspective rather than a flaw** in the OS-Map benchmark's utility.

We hope this summary assists the ACs in forming a comprehensive view of the paper's strong support and the nature of the remaining discussions. We respectfully hope that you consider the discussion and our responses, and the resolved concerns when making your recommendation.

Best regards,

Authors of 9576

---

### Meta-Review · Area_Chair_2M5n · 2026-01-06

**Summary:**

This paper introduces OS-MAP, a benchmark for evaluating computer-using agents, consisting of 416 tasks across 15 Ubuntu applications, organized along two axes: Automation Level (L1–L5) and Generalization Scope (S1–S3), with experiments showing that current agents achieve a low overall success rate (reported as ~11.5%) and near-zero performance on higher automation levels.

Across reviews, there is agreement that the benchmark artifact (tasks + environment + baselines) is substantial, and some reviewers value the paper’s organization and evaluation breadth. However, multiple reviewers raise core concerns about the conceptual validity and interpretability of the proposed evaluation framework, particularly the L-axis automation taxonomy, and these concerns remain insufficiently resolved after rebuttal and discussion. In addition, the Generalization Scope (S-axis) also raises concerns about practical interpretability and evaluability. Given that the paper’s central contribution is the benchmark *and its organizing evaluation framework*, the unresolved conceptual issues materially weaken the submission.

**Reviewer Concerns:**

### Concerns addressed by the rebuttal

* **Reviewer XYdi**

  * Concerns about task curation details being relegated to the appendix were addressed by the authors’ commitment to move and integrate these details into the main paper.
  * Issues with the framing of “proactivity” and “general” as higher-value regions were acknowledged, with agreement to revise this to “more challenging regions.”
  * After discussion, the reviewer explicitly stated that their concerns were cleared and raised their score from 4 to 6.

* **Reviewer VFWi (partially addressed)**

  * The authors provided justification for binary (0/1) end-state evaluation and explained why process-based metrics are difficult in open computer-use environments.
  * The manual cost of task construction and the hybrid human–LLM task creation pipeline were clarified.
  * Scope limitations regarding L5 tasks, task-generation automation, and cross-platform support were clearly delineated.
  * The reviewer acknowledged that these explanations addressed several of their initial concerns.

### Concerns still outstanding

* **Reviewer vHki (core unresolved issues)**

  * The reviewer consistently maintains that the **Automation Level (L-axis) conflates agent autonomy with task complexity**, limiting interpretability and making it unclear what capabilities are actually being measured.
  * The rebuttal’s argument that autonomy and complexity are “interdependent” is explicitly stated by the reviewer as insufficient to resolve the conceptual ambiguity.
  * The reviewer also maintains that the **ladder-like, sequential structure of L-levels** imposes an unrealistic progression that does not reflect parallel or orthogonal agent skills.
  * Despite multiple rebuttal rounds, the reviewer explicitly states that these conceptual concerns remain unresolved and that they are maintaining their original score.

* **Reviewer VFWi (remaining limitations)**

  * Even after rebuttal, the reviewer continues to view the benchmark as primarily focused on **episodic, single-round execution**, without directly evaluating persistent or multi-round interaction.
  * The reviewer also notes remaining room for more explicitly semantic-heavy tasks.
  * These issues are acknowledged by the authors as future work rather than resolved in the current submission.

**Reviewer Scores:**

* **Reviewer vHki**
  Explicitly stated multiple times that the rebuttal does **not resolve the core conceptual concerns** and that they are **maintaining their original score**. The score would **remain at 4**.

* **Reviewer VFWi**
  Acknowledged that several concerns were addressed but explicitly stated that **remaining limitations still affect their overall view** of the current version. The reviewer did not indicate a score increase; thus the score would **remain at 4**.

* **Reviewer bYpX**
  Strongly positive throughout, identified no weaknesses, and did not raise any questions. The score would **remain unchanged at 10**.

* **Reviewer XYdi**
  Explicitly stated that their concerns were resolved after discussion and **raised the score from 4 to 6**, which already reflects the post-discussion outcome.

---

### Decision · Program_Chairs · 2026-01-26

Reject